**Data Availability Statement:** All relevant data is contained in the manuscript.

**Funding:** DHS STITEL17GO Cystic Fibrosis Foundation https://www.cff.org/ The funders had

# Surface conjugation of antibodies improves nanoparticle uptake in bronchial epithelial cells

Valerie L. Luks[1,2], Hanna Mandl[3], Jenna DiRito🄸[1,3], Christina Barone[4], Mollie R. Freedman-Weiss[1], Adele S. Ricciardi[1,2], Gregory G. Tietjen[1,3], Marie E. Egan[4], W. Mark Saltzman[3], David H. Stitelman🄸[1] *

1 Department of Surgery, Yale University, New Haven, CT, United States of America, 2 Department of Surgery, University of Pennsylvania, Philadelphia, PA, United States of America, 3 Department of Biomedical Engineering, Yale University, New Haven, CT, United States of America, 4 Department of Pediatrics, Yale University, New Haven, CT, United States of America

* david.stitelman@yale.edu

## Abstract

### Background

Advances in Molecular Therapy have made gene editing through systemic or topical administration of reagents a feasible strategy to treat genetic diseases in a rational manner. Encapsulation of therapeutic agents in nanoparticles can improve intracellular delivery of therapeutic agents, provided that the nanoparticles are efficiently taken up within the target cells. In prior work we had established proof-of-principle that nanoparticles carrying gene editing reagents can mediate site-specific gene editing in fetal and adult animals in vivo that results in functional disease improvement in rodent models of β-thalassemia and cystic fibrosis. Modification of the surface of nanoparticles to include targeting molecules (e.g. antibodies) holds the promise of improving cellular uptake and specific cellular binding.

### Methods and findings

To improve particle uptake for diseases of the airway, like cystic fibrosis, our group tested the impact of nanoparticle surface modification with cell surface marker antibodies on uptake in human bronchial epithelial cells in vitro. Binding kinetics of antibodies (Podoplanin, Muc 1, Surfactant Protein C, and Intracellular Adhesion Molecule-1 (ICAM)) were determined to select appropriate antibodies for cellular targeting. The best target-specific antibody among those screened was ICAM antibody. Surface conjugation of nanoparticles with antibodies against ICAM improved cellular uptake in bronchial epithelial cells up to 24-fold.

### Conclusions

This is a first demonstration of improved nanoparticle uptake in epithelial cells using conjugation of target specific antibodies. Improved binding, uptake or specificity of particles delivered systemically or to the luminal surface of the airway would potentially improve efficacy, reduce the necessary dose and thus safety of administered therapeutic agents. Incremental

no role in study design, data collection and analysis, decision to publish, or preparation of the manuscript. WMS UG3- HL147352 National Institutes of Health https://www.nih.gov/institutes-nih The funders had no role in study design, data collection and analysis, decision to publish, or preparation of the manuscript.

**Competing interests:** The authors have declared that no competing interests exist.

improvement in the efficacy and safety of particle-based therapeutic strategies may allow genetic diseases such as cystic fibrosis to be cured on a fundamental genetic level before birth or shortly after birth.

## Introduction

Six percent of total births worldwide are complicated by birth defects and a large portion of these are the result of a single-gene mutation [1–4]. Cystic fibrosis (CF) is one such monogenic disease. With an incidence in the United States of one in every 3500 births and a worldwide prevalence of over 70,000 this complex disease is the second most common lethal inherited disorder [3, 5–7]. The most common genotypic variant of Cystic Fibrosis is a three base-pair deletion (delF508) in the cystic fibrosis transmembrane conductance regulator (CFTR) gene causing no expression of this channel on the cell's surface [6, 8]. One in every 25 people, in the United States carry a mutated copy of the CFTR gene [5]. Despite rigorous characterization, CF remains incurable. Therapy instead revolves around symptom management, which, due to increasingly sophisticated treatment, is often successful—though not without significant life-style impact.

Given the pathophysiology of diseases like CF, gene editing represents the ideal approach for curative gene correction [9]. Gene editing leaves gene products safely under native regulation and allows for permanent and exponential correction of cells and their progeny [10, 11]. However, there exist several barriers to successful gene editing. To successfully deliver any therapeutic agent to the desired cell or tissue type *in vivo*, they must be stably packaged. To this end, nanoparticles (NP)s have been used as stable and modifiable vessels to deliver drugs and small molecules to cells [12–16]. Delivering therapeutic agents via NPs can allow for increased circulation time of the agents and a decreased overall required dose [12, 13]. Polymers such as poly (lactic acid) (PLA) and poly-lactic-co-glycolic acid (PLGA) are approved by the FDA and are especially useful for therapeutic delivery due to their biocompatibility, biodegradability, and the ability to regulate the controlled release of contents over time [17–19].

The ability to package non-nucleases-based editing constructs within these NPs is vital for intracellular delivery; loaded NPs have now been shown to safely and efficiently mediate gene correction in human cells *in vitro* and in animal models *in vivo* [10, 11, 20–24]. Prior studies from our group have shown that when NPs loaded with gene editing agents are delivered trans-nasally to adult mice with CF there is a 1% level of editing in lung tissues—demonstrating the *in vivo* potential for gene editing, and a base upon which we can improve this technique [25].

Genetic diseases such as CF can be diagnosed early in fetal development. Currently however, despite these advances allowing for early fetal diagnosis, there remain no prenatal treatment options. The theoretical benefits of treating diseases such as CF prenatally is multifold. Treating diseases with progressive and systemic symptoms before phenotypic manifestation could render a fetal patient cured and asymptomatic for their lifetime. Also, a fetal patient, in comparison to a post-natal patient, is significantly smaller and composed of a proportionally greater number of stem and progenitor cells. In the case of gene editing, the impact after a relatively small dose, can be exponentially potent in the stem cells of the brain, skin, muscle, liver, and blood owing to their massive proliferation and migration [26–29]. Further, the fetus is immunotolerant based on its immature immune system. A recent study from our group took advantage of the benefits of the fetal host and demonstrated the efficacy of intrauterine fetal

gene editing via NP delivery. In this study, NPs loaded with gene editing agents delivered to fetuses with beta-thalassemia, another monogenic disease, intravenously, there was on average 6% gene editing frequency of total bone marrow cells without any off-target effects [20].

There remains however, a significant need for improvement in the uptake of these NPs. The uptake of NPs by various cell types is predominantly mediated by endocytosis, with subsequent escape of the internalized nanoparticle from the endosome [18, 30]. The efficiency of NP uptake is dependent on particle size, shape, charge, and surface properties—which are modifiable via proteins, receptor ligands, and cell-specific antibodies [18, 31–39]. Modifications to the surface of NPs that result in increased cell-contact time may indeed increase the rate of NP internalization and ultimately gene correction. Conjugation of specific antibodies to NPs has a dual benefit of both prolonging cell surface binding, thus improving NP uptake, and specifying the cell-type to which a NP binds and forms an endosome [39–41]. Polymers added to the NP, such as polyethylene glycol (PEG), allow construction of NPs with conjugated antibody on the surface, a method that is already approved for use in clinical trials. Antibody bound NPs thus presents a safe, biocompatible, and targetable method of delivering gene editing (or other therapeutic agents) to cells and tissues [39, 42, 43].

In this study, we hypothesized that NP uptake in bronchial epithelial cells can be increased by modifying the NP surface with cell-specific antibodies. Given our experience with prenatal and postnatal gene editing using reagents carried in NPs [20, 25], our hope is that this incremental improvement can yield higher cellular uptake of reagents with intent of phenotypic cure for monogenic diseases such as CF.

## Materials and methods

### Cell culture

Human CF bronchial epithelial cells (CFBE) (an immortalized cell line homozygous for the F508 deletion mutation) were cultured at 37˚C in growth media (1% L-glutamine, 1% penicillin-streptomycin, 10% fetal bovine serum in 1x EMEM by Mediatech Inc) on cell culture places that were pretreated with 0.3 mg/ml rat collagen in sterile 0.1% acetic acid. Cells were grown to confluence and passaged accordingly. For experimental purposes, cells were seeded onto collagen-treated 48-well or 96- well plates. Confluent cells in flask were washed with PBS, treated with TrypLE Express trypsin, and allowed to incubate for 10 minutes at 37˚C. Once detached, cells were centrifuged and resuspended in fresh media. To count cells, 1:1 aliquot of cells to dye was used. Equal concentrations of cell suspension were then seeded into each well. Cells adhered and grew for 48 hours before experimental use.

### Target selection and antibody testing

Appropriate targets were selected by literature review and with aid from the LungGENS database [44]. Selection criteria required that the targets be cell-surface markers, present on epithelial cells, and present in pulmonary tissues. Five antibodies against selected target antigens, which included: surfactant protein C (SFTPC) (Abcam ab90716), podoplanin (PDPN) (Abcam ab10288), mucin 1 (MUC1) (Abcam Ab70475), a second clone of mucin 1 (CD227) (BD Biosciences: Cat#550486), and intracellular adhesion molecule 1 (ICAM) (Biolegend Cat # 322703) were obtained. For each antibody, an appropriate isotype control antibody was studied as well: Rabbit IgG Isotype (for SFTPC) (Abcam ab27478) and Mouse IgG Isotype (PDPN, Muc 1 & ICAM) (BioLegend Cat#401404). To optimize targeting efficacy, three antigen properties were analyzed: accessibility, receptor density, and antigen-antibody affinity [45].

Accessibility of the antigens was analyzed by immunofluorescent staining of CFBE cells to confirm expression of the target. Cells were fixed with 4% paraformaldehyde (PFA) for 15

                                                                                                             

minutes at room temperature and then washed with PBS. Samples were blocked with FBS for one hour at room temperature and then incubated with primary antibody at 1:100 overnight at 4°C. Cells were washed and then exposed to appropriate species of secondary antibody conjugated to Alexa Fluor 488 fluorophore (Ex = 490nm, Em = 525nm) for 3 hours at room temperature. After final washing step, cells were counterstained with DAPI and mounted for analysis by confocal microscopy. Confocal imaging was performed on a Zeiss Cell Observer SD microscope.

### Receptor density and antibody binding analysis

Receptor density and antibody-antigen affinity were determined by antibody titration curves [39, 46]. Cells in 96-well plates were treated in triplicate with 60 μl of media containing primary antibody in concentrations ranging from 0.1 nM to 100 nM (experiments were performed in triplicate). The corresponding isotype antibody was used as a control for each antibody species. The cells were incubated for one hour at 37°C. Cells were washed with media, treated with 30nM concentration of secondary antibody conjugated to Alexa Fluor 488 fluorophore in cell culture media, and incubated again for one hour at 37°C. Cells were washed and trypsonized with 0.25% trypsin. Cold fluorescence activated cell sorting (FACS) buffer (2% fetal bovine serum in sterile PBS) was added to each well and cells were collected in individual FACS tubes and kept on ice for analysis. FACS was performed on an Attune NxT Flow Cytometer by Invitrogen. Untreated cells were used to determine forward- and side-scatter voltages and background fluorescence was subtracted from all experimental groups. FlowJo software was used for FACS analysis. Intensity data generated was used to create a non-linear one-site specific binding curve using Prism software by GraphPad. The calculated $B_{max}$ value, where the brightness plateaus, was used to approximate the receptor density. The calculated equilibrium dissociation constant ($K_d$), the concentration where half of peak brightness is attained, was used to approximate the antibody-antigen affinity [46].

### Nanoparticle synthesis, characterization and antibody conjugation

Fluorescent PLA- PEG NPs were formulated by the NanoAssemblr Benchtop system by Precision Nanosystems. Polylactic acid–polyethylene glycol (PLA-PEG) polymer was 16–5 kDal MW (PolySciTech). By altering the starting concentration of the polymer, the diameter of the nanoparticles can be consistently altered. For PLA-PEG NPs, a starting concentration of 40 mg/ml in 75% DMSO resulted in 150nm particles. Green fluorescent fluorescent dye (DiO) at 0.5% w/w was dissolved in 10% DMSO. The polymer-dye mixture was injected through the organic inflow channel of the microfluidic mixer and a 2% PVA in water solution was injected through the aqueous inflow channel of the microfluidic mixer. The NPs were created with a 1:1 flow rate of aqueous to organic solvents, 8 ml/min total flow rate, and 2 ml total volume. NPs were collected in water. Solvent was removed by centrifugation and washing. NP batches were concentrated by a factor of 10 and dialyzed 10 times. NPs were then frozen at -80°C and lyophilized for 48 hours in preweighed Eppendorf tubes. Final weight of dehydrated NPs was noted, and NPs were stored at -20°C. NP size was determined by DLS, and a standard curve was produced (Fig 3A).

Antibody conjugation to PLA-PEG NPs occurred via 1-ethyl-3-(3- dimethylaminopropyl) carbodiimide hydrochloride (EDC) mediated carboxyl-amine crosslinking. Antibodies containing sodium azide or other preservatives were dialyzed overnight to remove any molecules that could inhibit coupling. The antibody concentration was adjusted to a working volume of 0.55 mg/ml. The conjugation reaction occurs in three steps: activation, conjugation, and quenching. 15 An aliquot of NPs at 5mg/ml was thawed and sonicated to resuspend the

particles. To activate the NPs, 57 μl of 1 M MES buffer (pH 5.5) was added under vortex. Next, 57 μl of 100 mg/ml sulfo-(N-hydroxysulfosuccinimide) (NHS) was added under vortex to stabilize the reaction. Finally, 57 μl of EDC was added under vortex to activate the carboxyl group on the PEG groups. The NPs were vortexed vigorously for 15 minutes to complete the activation. For the conjugation phase of the reaction, the NPs were split into two aliquots and micro-centrifuged at 21,500g for 10 minutes. The antibody was prepared by adding 1 M MES to a final concentration of 50 mM. Once the centrifugation of the NPs was complete, the supernatant was carefully aspirated, and the activated NPs were resuspended in 50 μl of 50 mM MES in PBS. The activated NPs were then added to the prepared antibody solution under mild vortex. The NPs were vortexed vigorously for one hour to complete the conjugation. To quench the reaction, 6 $\mu$l of 1 M Tris buffer (pH 9) was added to the conjugated NP solution. The NPs were then micro-centrifuged at 21,500g for 10 minutes and the supernatant was carefully aspirated. The conjugated NPs were resuspended in PBS and diluted to desired concentration. Aliquots were snap frozen in liquid nitrogen and stored at -80˚C [16].

### In vitro nanoparticle treatments

**Conjugated nanoparticle treatment.** CFBE cells were seeded in 48-well plates. Each batch of antibody-conjugated NPs was tested at two concentrations: 25 μg/ml and 50 μg/ml in 500 μl total volume of cell culture media and tested in quadruplicate. Isotype-conjugated NPs were used as matched controls for each experimental antibody. Cells were incubated with the conjugated NPs for 2 hours at 37˚C. The cells were subsequently washed six times with warm FACS buffer before being trypsonized with 0.25% trypsin. Cold FACS buffer was added to each well and cells were collected in individual FACS tubes and kept on ice for analysis. Flow cytometry was performed on an Attune NxT Flow Cytometer by Invitrogen. Untreated cells were used to determine forward- and side-scatter voltages and background fluorescence was subtracted from all experimental groups. FlowJo software was used for FACS analysis for mean fluorescence intensity of cells that took up fluorescent loaded NPs. Samples were compared by student t test with p<0.01 considered significant.

## Results

### Antigen accessibility and density in CFBE cells

Human CFBE cells were stained with 5 antibody/epitopes predicted to be prominent of the cell surface of CFBE cells: PDPN, MUC1, CD227 (a second Muc 1 antibody), ICAM, with a secondary green fluorescent antibody (AlexaFlour488), and a DAPI nuclear stain. These targets were selected based on literature review of proteins that were expressed on the luminal surface of lung epithelium. When analyzed with confocal microscopy, a high density of ICAM was demonstrated (Fig 1).

Antibody titration curves for the candidate surface markers were created with CFBE cells to determine receptor density and antibody affinity properties. Muc1 and ICAM (Fig 2C–2E) fluorescent intensity increased in a dose-dependent manner, while increasing concentrations of anti-PDPN and anti-SFTPC antibodies did not modulate fluorescent signals (Fig 2A and 2B). At 30nM, a representative intermediate concentration, comparison of antibody binding to isotype is show in Fig 2F, demonstrating right-shifted Muc1, ICAM, and CD227 antibody curves compared to the isotype curve—with ICAM having the greatest fluorescent intensity.

When fit to a specific binding curve, the $B_{max}$ and $K_d$ values can be calculated for MUC1, CD227, and ICAM (Fig 2C–2E). These values are representative of the relative receptor density and antigen-antibody affinity respectively. ICAM had the greatest receptor density with a $B_{max}$ of 32,161 (vs Muc1 $B_{max}$ = 4940, CD227 $B_{max}$ = 5862). It is expected that receptor density

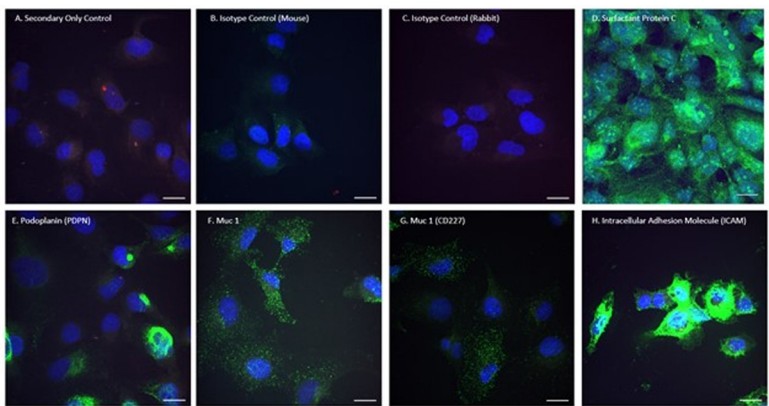

**Fig 1.** CFBE Cells that were A) Unstained B&C) Mouse and Rabbit Isotype controls and stained for: D) Surfactant Protein C, E) Podoplanin, F&G) Two different Muc 1 (CD227) antibody clones, and H) Intracellular Adhesion Molecule 1 (ICAM). 40X Confocal image. (Blue = DAPI nuclear stain) (Bar = 20 micron).

would be comparable for Muc 1 using two different antibodies. The tested clone of antibody against ICAM demonstrated the greatest binding affinity with the lowest $K_d$ of 0.092 nM (vs MUC1 = 2.414, CD227 = 1.584). SFTPC and PDPN results did not fit a curve, thus $B_{max}$ and Kd calculations were not possible (Fig 2A and 2B).

## Antibody conjugation to nanoparticles influences bronchial epithelial cell uptake

Synthesized NPs made from PLA-PEG were characterized by SEM (Fig 3A), which confirmed consistent ability to create spherical NPs of desired size and distribution. Green fluorescent,

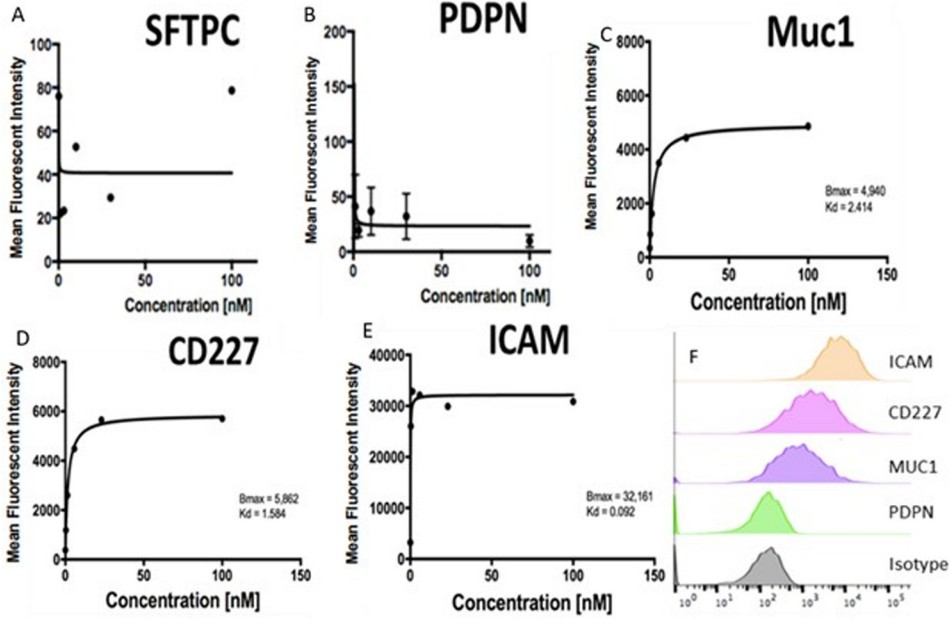

**Fig 2. Antibody Binding Curves of antibody concentration (x axis) versus Mean Fluorescence Intensity (MFI) (y axis) demonstrating receptor density ($B_{max}$) and antibody-antigen affinity ($K_d$).** A) Surfactant Protein C (SFTPC) B) Podoplanin (PDPN) (SFTPC and PDPN did not fit a curve, thus $B_{max}$ and Kd calculations were not possible). C) Muc 1 D) CD 227 (second clone of Muc 1 antibody) E) Intracellular Adhesion Molecule 1 (ICAM). F) Comparison of MFI distribution at an antibody concentration of 30 nM of each antibody tested.

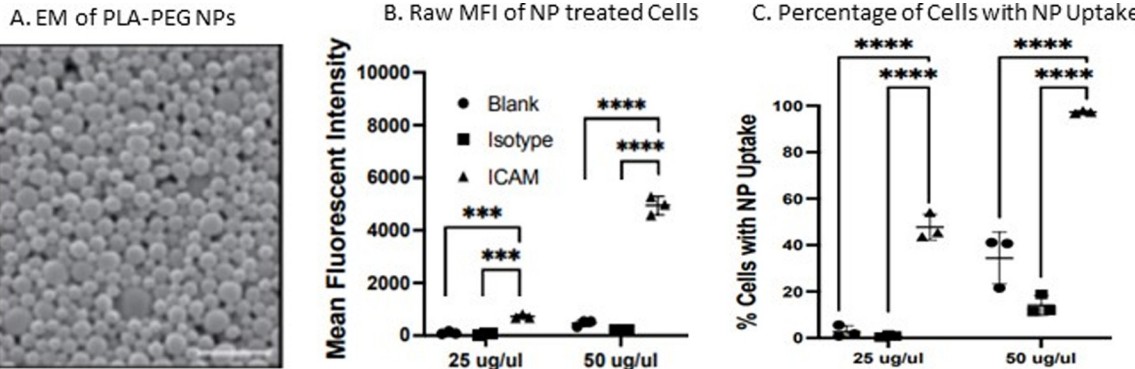

**Fig 3. Nanoparticles synthesis with surface conjugation to antibody improves uptake in CFBE cells in vitro.** A) By SEM, PLA-PEG particles are uniform in size (Bar = 1μm). B&C) Flow cytometry of CFBE Cells treated with green (DiO) dye. These NPs were either, blank unconjugated, isotype conjugated or ICAM antibody conjugated. B MFI of treated cells and C) percentage cells with NP internalization were measured. (n = 3 for each treatment) (*** p<0.001 & **** p<0.0001).

DiO-loaded PLA-PEG NPs were conjugated with ICAM antibodies (and isotype control) and were subsequently treated on CFBE cells there was a clear antibody-dependent increased uptake of NPs compared to unmodified NPs and NPs conjugated to isotype antibodies (Fig 3B & 3C). MFI measures brightness of cells, so the more NPs that are internalized, the higher the MFI. Antibody conjugation with ICAM increases MFI by 12-fold at the lower dose tested and 24-fold at the higher dose tested (Fig 3B). By flow cytometry, the number of treated cells that had NP uptake was drastically improved by ICAM antibody conjugation. At the lower dose, with ICAM conjugation, nearly 50% of cells had NP uptake compared to less than one percent for isotype conjugated antibody (60-fold increase, p<0.0001). At the higher dose tested, ICAM conjugation resulted in uptake in 98% of cells, compared to 14% for the isotype control (7-fold increase, p<0.0001) (Fig 3C).

## Discussion

Impacting over 70,000 people worldwide, CF represents a common disease with lifetime morbidity, which is theoretically curable by gene editing even before symptom onset. NP delivery of therapeutic gene-editing agents represents a vast and promising new technique for CF treatment clinically both prenatally and postnatally. Already, NP-based gene editing therapy has proven efficacious in mouse-models of monogenic diseases before and after birth [10, 11, 22, 25, 30]. Further, with the knowledge that the fetal host is smaller, more immunotolerant, and has a greater proportion of multipotent stem-cells which undergo massive proliferation and migration—the fetus has been a promising target for nanoparticle-based gene therapy [10, 20]. Perhaps most compelling for NP-based fetal gene therapy, is that given an early fetus is asymptomatic from their genetic disease while *in-utero*, gene editing at this stage could offer disease cure prior even to phenotypic onset [9, 26–29].

For lung-based monogenic diseases, such as CF, the apical epithelial surface of the lung is the main target for NP therapy [25]. The route that in principle, delivers the greatest therapeutic dose to the epithelial lung cells in the most specific manner, would be inhalation. Postnatal administration of particles is technically feasible but needs to overcome the mucosal barrier to uptake into cells. Prenatal administration (intra-amniotic or intra-tracheal) are also technically feasible. We thus sought to improve the delivery of therapeutic agents with modified NPs, to improve the NP uptake and thus efficiency of gene editing.

By conjugating tissue-specific, cell-surface antibodies to NPs, a larger portion of NPs are internalized by their specific target cells—a concept we hoped to apply to the rodent model [39]. In this study, ICAM was found on immunohistochemistry and antibody-binding curves, to be the lung cell-surface target with both the greatest cell surface density and the greatest affinity for its antibody. ICAM is a versatile and attractive target antibody for several reasons. Found on endothelial surfaces as well as respiratory epithelium, conjugated nanoparticles may have increased internalization when delivered to the luminal surface of the lung. Further, ICAM is upregulated during inflammatory states in pulmonary diseases, such as CF—perhaps further increasing its surface density and utility as a target [47]. The selection of targets for these antibodies is complex. The target must be on the cell surface that is exposed to the route of delivery, have reasonable cell surface density (high $B_{max}$), and strong antibody-target affinity (low $K_d$). We selected ICAM because it had the best performance characteristics in Vivo but PDPN and Muc 1 remain possible targets in the fetal model. Although SFPC was an attractive target based on cell staining (Fig 1) and LungGens mRNA expression data [44], much of this protein is likely inside the cell and not a useful cell surface target.

The conjugation of ICAM antibody to the surface of these NPs appears to drastically increase the amount of NPs that are taken up by cells in vitro demonstrated by the 12–24 fold increase in MFI in cells treated with ICAM antibody conjugated NPs compared to control NPs (Fig 3B). This method also appears to increase the number of cells where NP uptake is successful. At the higher dose tested 98% of cells had NP uptake compared to 14% in the control group (Fig 3C). Antibody conjugation holds the promise of being able to use a lower dose on NP, which may improve safety if a fraction of the dose were needed. The improvement in the number of cells with successful uptake may also improve therapeutic efficacy.

There are several limitations to this study—the CFBE cells that were used and upon which the optimal antibody (ICAM) was selected were of human origin, while our in vivo models are murine-based so to test these agents in vivo a new set of particles need to be synthesized. Selection of the optimal target may be different in postnatal and fetal tissue. Finally, increased uptake of particles, improves the intracellular dose, but the unpackaging of nanoparticles intracellularly and success of editing reagents is a complex process. Further studies are necessary to establish if increase uptake of particles loaded with editing reagents results in improved levels of editing.

In summary, we demonstrated that conjugating tissue-specific, cell-surface antibodies to NPs allowed greater NP delivery and internalization in CFBE cells in vitro. Specifically, ICAM-conjugated PLA-PEG NPs were internalized with significantly greater frequency than unmodified, or isotype conjugated NPs. The next step in this study is clearly to conjugate mouse antibodies to NPs and test their performance in terms of uptake and editing in postnatal and fetal mice. Our hope is that incremental improvements in the efficacy and specificity of particle-based gene editing will carry the potential to reduce the necessary systemic dose and increase the efficiency of gene-editing of lung-based monogenic diseases such as CF.

## Author Contributions

**Conceptualization:** Valerie L. Luks, Adele S. Ricciardi, Gregory G. Tietjen, Marie E. Egan, W. Mark Saltzman, David H. Stitelman.

**Data curation:** Valerie L. Luks, Jenna DiRito, David H. Stitelman.

**Formal analysis:** Valerie L. Luks, Marie E. Egan, W. Mark Saltzman, David H. Stitelman.

**Funding acquisition:** W. Mark Saltzman, David H. Stitelman.

**Investigation:** Valerie L. Luks, Hanna Mandl, Jenna DiRito, Christina Barone, Mollie R. Freedman-Weiss, David H. Stitelman.

**Methodology:** Valerie L. Luks, Hanna Mandl, Jenna DiRito, W. Mark Saltzman, David H. Stitelman.

**Project administration:** David H. Stitelman.

**Resources:** David H. Stitelman.

**Supervision:** Marie E. Egan, David H. Stitelman.

**Writing – original draft:** Valerie L. Luks.

**Writing – review & editing:** Adele S. Ricciardi, Gregory G. Tietjen, W. Mark Saltzman, David H. Stitelman.

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
