## [Decision Letter · Decision Letter 0]

28 Jan 2022

PONE-D-21-38727“Surface conjugation of Antibodies Improves Nanoparticle Uptake in Bronchial Epithelial Cells,”PLOS ONE

Dear Dr. Stitelman,

Thank you for submitting your manuscript to PLOS ONE. After careful consideration, we feel that it has merit but does not fully meet PLOS ONE’s publication criteria as it currently stands. Therefore, we invite you to submit a revised version of the manuscript that addresses the points raised during the review process.

Considering the emphasis of the manuscript on using nanoparticles as a tool to treat CF cells, some experimental measure of functional efficacy beyond uptake is needed. The suggestions of adding images and individual data points for Figure 3 should also be addressed.  Methodology related to uptake quantitation should be assessed and there are several places where the manuscript would benefit from additional editing.

We look forward to receiving your revised manuscript.

Kind regards,

Michael Koval

Academic Editor

PLOS ONE

Journal Requirements:

2. "We note that the grant information you provided in the ‘Funding Information’ and ‘Financial Disclosure’ sections do not match. 

[This work was supported by the National Institutes of Health (UG3- HL147352 to WMS). VLL was supported by a Cystic Fibrosis Foundation award (STITEL17G0).]

[DHS STITEL17GO Cystic Fibrosis Foundation

https://www.cff.org/

WMS UG3- HL147352 National Institutes of Health

https://www.nih.gov/institutes-nih

The funders had no role in study design, data collection and analysis, decision to publish, or preparation of the manuscript]

5. PLOS requires an ORCID iD for the corresponding author in Editorial Manager on papers submitted after December 6th, 2016. Please ensure that you have an ORCID iD and that it is validated in Editorial Manager. To do this, go to ‘Update my Information’ (in the upper left-hand corner of the main menu), and click on the Fetch/Validate link next to the ORCID field. This will take you to the ORCID site and allow you to create a new iD or authenticate a pre-existing iD in Editorial Manager. Please see the following video for instructions on linking an ORCID iD to your Editorial Manager account: https://www.youtube.com/watch?v=_xcclfuvtxQ.

Reviewers' comments:

Reviewer's Responses to Questions

**Comments to the Author**

1. Is the manuscript technically sound, and do the data support the conclusions?

Reviewer #1: Yes

2. Has the statistical analysis been performed appropriately and rigorously? 

Reviewer #1: I Don't Know

3. Have the authors made all data underlying the findings in their manuscript fully available?

Reviewer #1: Yes

4. Is the manuscript presented in an intelligible fashion and written in standard English?

Reviewer #1: Yes

5. Review Comments to the Author

Reviewer #1: Comments to the Author

The manuscript by Luks et al. describes the effect of conjugation of antibodies against surface proteins on the uptake of nanoparticles (NPs) in vitro using the human cystic fibrosis bronchial epithelial (CFBE) cells. The authors report increased NP uptake in the CFBE cells when the NPs are modified by cell surface protein antibodies. They choose several potential candidates for surface conjugation of the NPs. By analyzing the antibody binding kinetics, receptor density and antibody affinity are compared, and antibodies against the Intracellular Adhesion Molecule-1 (ICAM) are selected for further evaluation. The authors provide evidence that surface conjugation of NPs with antibodies against ICAM increases the uptake of such NPs by the CFBE cells. The findings of this study shed light on a potential approach to improve the delivery of therapeutic reagents loaded by the aforementioned NPs. There are no major ethical or methodological concerns. The manuscript is concise and well written. However, the quantity of data is not adequate as a candidate for an original research article. Instead, it would be more appropriate to be considered as a short communication or methodology report. Nevertheless, there are some comments that would help to improve the manuscript.

Major points:

1. Based on the description of the conjugated nanoparticle treatment on page 9, the authors analyzed the mean fluorescence intensity of “cells that took up fluorescent loaded NPs”. How was the percentage of cells that took up the NPs?

2. The authors should consider including images for the CFBE cells that internalized NPs for the data shown in Figure 3C.

3. Individual data points should be presented in Figure 3A and 3C.

4. The authors used CFBE cells, which carry the F508 deletion mutation, as the in vitro model in this study, and described the significance of improving the approaches used for molecular therapy for cystic fibrosis to a great extent. Have the authors observed any effect(s) using the NPs loaded with gene editing agents? Will the improved NP uptake benefit the outcomes/functional readout of gene editing in the CFBE cells?

Minor points:

1. It might help the readers if the rationale of target selection on page 6 is explained at the beginning of the Results session.

2. Labeling and units for the y-axes are missing in Figure 2 and Figure 3C.

3. The font used for “Greek letter mu (µ)” symbols needs to be standard and unified throughout the manuscript.

4. Page 12. Line 3, what is “DiO” short for?

6. PLOS authors have the option to publish the peer review history of their article (what does this mean?). If published, this will include your full peer review and any attached files.

Reviewer #1: No

---

## [Author Response · Author response to Decision Letter 0]

13 Mar 2022

Please See "Response to Reviewers" Document

---

## [Decision Letter · Decision Letter 1]

17 Mar 2022

“Surface conjugation of Antibodies Improves Nanoparticle Uptake in Bronchial Epithelial Cells,”

PONE-D-21-38727R1

Dear Dr. Stitelman,

We’re pleased to inform you that your manuscript has been judged scientifically suitable for publication and will be formally accepted for publication once it meets all outstanding technical requirements.

Kind regards,

Michael Koval

Academic Editor

PLOS ONE

Additional Editor Comments (optional):

Reviewers' comments:

Reviewer's Responses to Questions

**Comments to the Author**

1. If the authors have adequately addressed your comments raised in a previous round of review and you feel that this manuscript is now acceptable for publication, you may indicate that here to bypass the “Comments to the Author” section, enter your conflict of interest statement in the “Confidential to Editor” section, and submit your "Accept" recommendation.

Reviewer #1: All comments have been addressed

2. Is the manuscript technically sound, and do the data support the conclusions?

Reviewer #1: Yes

3. Has the statistical analysis been performed appropriately and rigorously? 

Reviewer #1: I Don't Know

4. Have the authors made all data underlying the findings in their manuscript fully available?

Reviewer #1: Yes

5. Is the manuscript presented in an intelligible fashion and written in standard English?

Reviewer #1: Yes

6. Review Comments to the Author

Reviewer #1: Comments to the Author

In this revised manuscript, Luks et al. have been responsive to the most of the previous comments. Please see below.

Major points:

1. Based on the description of the conjugated nanoparticle treatment on page 9, the authors analyzed the mean fluorescence intensity of “cells that took up fluorescent loaded NPs”. How was the percentage of cells that took up the NPs?

Response: Our collaborators have classically looked at mean fluorescence intensity to see if more particles were going into each cell. We had not analyzed the data looking at cell number with particle uptake previously and were impressed by these results. Upon review of the flow cytometry data, we see that at our low dose, the conjugated particles enter over 50% of cells compared to less than a percent of cells (60 fold increase) and with the higher dose, conjugated particles enter about 98% of cells compared to 14% of cells (7 fold increase). This has been added to the figures, results and discussion.

Reviewer’s comment: Authors’ response to this comment is satisfactory.

2. The authors should consider including images for the CFBE cells that internalized NPs for the data shown in Figure 3C.

Response: We have no more of these particles and do not have these images. We feel the flow cytometry data is quantitative and the flow cytometry analysis has been improved.

Reviewer’s comment: Authors failed to address this comment because of technical issues, however the added flow cytometry data is helpful.

3. Individual data points should be presented in Figure 3A and 3C.

Response: We have removed figure 3A to make room for what are now figure 3B and 3C. We have changed the graphs from bar graphs to graphs of individual data points.

Reviewer’s comment: Authors’ response to this comment is appropriate.

4. The authors used CFBE cells, which carry the F508 deletion mutation, as the in vitro model in this study, and described the significance of improving the approaches used for molecular therapy for cystic fibrosis to a great extent. Have the authors observed any effect(s) using the NPs loaded with gene editing agents? Will the improved NP uptake benefit the outcomes/functional readout of gene editing in the CFBE cells?

Response: The conjugation technique that we have used for this project results in the antibody to be oriented in random direction so the binding domain may be facing away from the particle (ideal) but more often aimed in the wrong direction. We have developed a new method of antibody conjugation that results in the antibodies aimed in the ideal direction and this new generation of particles enters cells more efficiently than the method described in this manuscript. The editing reagents are expensive and in limited supply and we will not be using them in antibody conjugated particles synthesized in the manner described in this paper. We feel that this demonstration of increased particle uptake using antibody conjugation is important and studies of intracellular release and therapeutic efficacy are a necessary next step.

Reviewer’s comment: Authors failed to address this comment because of another technical issue and practical reasons, which is unfortunate but understandable.

Minor points:

1. It might help the readers if the rationale of target selection on page 6 is explained at the beginning of the Results session.

Response: A sentence describing the rationale was added to the beginning of the result section on page 9.

Reviewer’s comment: The authors answered this comment, however the references used for the “literature review” should be listed.

2. Labeling and units for the y-axes are missing in Figure 2 and Figure 3C.

Response: The figures are now labeled accurately.

Reviewer’s comment: Response to this comment is appropriate.

2. The font used for “Greek letter mu (μ)” symbols need to be standard and unified throughout the manuscript.

Response: All “Greek letter mu” were changed to μ.

Reviewer’s comment: Response to this comment is adequate.

4. Page 12. Line 3, what is “DiO” short for?

Reviewer’s comment: Page 7, line 18, when explain “DiO”, there is a duplication of the word ‘florescent’.

7. PLOS authors have the option to publish the peer review history of their article (what does this mean?). If published, this will include your full peer review and any attached files.

Reviewer #1: No

---

## [Editor Report · Acceptance letter]

24 Mar 2022

PONE-D-21-38727R1 

Surface conjugation of Antibodies Improves Nanoparticle Uptake in Bronchial Epithelial Cells 

Dear Dr. Stitelman:

I'm pleased to inform you that your manuscript has been deemed suitable for publication in PLOS ONE. Congratulations! Your manuscript is now with our production department. 

Kind regards, 

on behalf of

Dr. Michael Koval 

Academic Editor

PLOS ONE